# Increased Citrullinated Histone H3 Levels in the Early Post-Resuscitative Period Are Associated with Poor Neurologic Function in Cardiac Arrest Survivors—A Prospective Observational Study

**DOI:** 10.3390/jcm8101568

**Published:** 2019-10-01

**Authors:** Lisa-Marie Mauracher, Nina Buchtele, Christian Schörgenhofer, Christoph Weiser, Harald Herkner, Anne Merrelaar, Alexander O. Spiel, Lena Hell, Cihan Ay, Ingrid Pabinger, Bernd Jilma, Michael Schwameis

**Affiliations:** 1Clinical Division of Hematology and Hemostaseology, Department of Medicine I, Medical University of Vienna, 1090 Vienna, Austria; lisa-marie.mauracher@meduniwien.ac.at (L.-M.M.); nina.buchtele@meduniwien.ac.at (N.B.); lena.hell@meduniwien.ac.at (L.H.); cihan.ay@meduniwien.ac.at (C.A.); ingrid.pabinger@meduniwien.ac.at (I.P.); 2Department of Clinical Pharmacology, Medical University of Vienna, 1090 Vienna, Austria; christian.schoergenhofer@meduniwien.ac.at (C.S.); bernd.jilma@meduniwien.ac.at (B.J.); 3Department of Emergency Medicine, Medical University of Vienna, 1090 Vienna, Austria; christoph.weiser@meduniwien.ac.at (C.W.); Harald.herkner@meduniwien.ac.at (H.H.); anne.merrelaar@meduniwien.ac.at (A.M.); alexander.spiel@meduniwien.ac.at (A.O.S.); 4I.M. Sechenov First Moscow State Medical University (Sechenov University), 119146 Moscow, Russia

**Keywords:** neutrophil extracellular traps, citrullinated histone H3, cardiac arrest, neurologic function

## Abstract

The exact contribution of neutrophils to post-resuscitative brain damage is unknown. We aimed to investigate whether neutrophil extracellular trap (NET) formation in the early phase after return of spontaneous circulation (ROSC) may be associated with poor 30 day neurologic function in cardiac arrest survivors. This study prospectively included adult (≥18 years) out-of-hospital cardiac arrest (OHCA) survivors with cardiac origin, who were subjected to targeted temperature management. Plasma levels of specific (citrullinated histone H3, H3Cit) and putative (cell-free DNA (cfDNA) and nucleosomes) biomarkers of NET formation were assessed at 0 and 12 h after admission. The primary outcome was neurologic function on day 30 after admission, which was assessed using the five-point cerebral performance category (CPC) score, classifying patients into good (CPC 1–2) or poor (CPC 3–5) neurologic function. The main variable of interest was the effect of H3Cit level quintiles at 12 h on 30 day neurologic function, assessed by logistic regression. The first quintile was used as a baseline reference. Results are given as crude odds ratio (OR) with 95% confidence interval (95% CI). Sixty-two patients (79% male, median age: 57 years) were enrolled. The odds of poor neurologic function increased linearly, with 0 h levels of cfNDA (crude OR 1.8, 95% CI: 1.2–2.7, *p* = 0.007) and nucleosomes (crude OR 1.7, 95% CI: 1.0–2.2, *p* = 0.049), as well as with 12 h levels of cfDNA (crude OR 1.6, 95% CI: 1.1–2.4, *p* = 0.024), nucleosomes (crude OR 1.7, 95% CI: 1.1–2.5, *p* = 0.020), and H3Cit (crude OR 1.6, 95% CI: 1.1–2.3, *p* = 0.029). Patients in the fourth (7.9, 95% CI: 1.1–56, *p* = 0.039) and fifth (9.0, 95% CI: 1.3–63, *p* = 0.027) H3Cit quintile had significantly higher odds of poor 30 day neurologic function compared to patients in the first quintile. Increased plasma levels of H3Cit, 12 h after admission, are associated with poor 30 day neurologic function in adult OHCA survivors, which may suggest a contribution of NET formation to post-resuscitative brain damage and therefore provide a therapeutic target in the future.

## 1. Introduction

In cardiac arrest survivors good neurologic outcome remains difficult to achieve [1]. Brain injury does not occur solely during circulatory interruption, but may progress during the reperfusion period after sustained return of spontaneous circulation (ROSC) [2]. Ischemic reperfusion is considered a main trigger of a complex cascade of pro-inflammatory and pro-thrombotic events occurring hours to days after resuscitation, which may impair cerebral microvascular perfusion despite restoration of macrovascular flow [3,4]. Recent data suggest that the response of neutrophils to hypoxia could be an early and critical mediator of ischemic reperfusion injury [5]. This is consistent with previous studies reporting substantial mortality and neurologic morbidity in resuscitated cardiac arrest patients with an elevated number of blood neutrophils in relation to other leukocyte counts [6,7,8]. The mechanisms by which neutrophils may contribute to post-resuscitative brain damage have, however, not yet been elucidated.

In recent years, neutrophil extracellular traps (NETs) have emerged as a central player in inflammation, thrombogenesis, and cardiovascular disease [9,10,11,12,13,14]. NETs are chromatin fibers consisting of histones, cell free DNA (cfDNA), and granular proteins, and are released within minutes [15] to hours [16] following activation by various stimuli including ischemia and reperfusion [17]. While NETs have primarily been recognized as mediators of antimicrobial host defense, they may exert detrimental inflammatory and procoagulant effects causing endothelial damage, platelet activation, microvessel occlusions, and ultimately tissue malperfusion [18]. In particular, neutrophil histones and DNA are considered cytotoxic and procoagulant components of NETs [19], and have been implicated in organ damage in various noninfectious conditions [20,21]. Citrullination of histone H3 (H3Cit) by peptidylarginine deiminase 4 (PAD4) is a key signal for chromatin decondensation and NET formation [22]. H3Cit is commonly accepted as a NET biomarker, and has been measured in various studies to investigate NET formation. Despite this, the role of NETs in cardiac arrest has not yet been investigated. Pro-inflammatory and pro-thrombotic properties, however, render them possible mediators of neutrophil-borne brain injury after successful resuscitation.

We hypothesized that neutrophil extracellular trap (NET) formation may be associated with poor neurologic function after cardiac arrest. This study aimed to assess plasma levels of NET biomarkers in the early phase after ROSC and investigate its association with short-term neurologic function in a selected cohort of out-of-hospital cardiac arrest (OHCA) survivors.

## 2. Methods

This prospective single-center observational cohort study was conducted at the Emergency Department at the Medical University of Vienna. Adult (≥18 years) OHCA survivors with cardiac origin who received in-hospital targeted temperature management (33 ± 0.5 °C) were enrolled. Exclusion criteria included current oral anticoagulation therapy, thrombolytic therapy, intravascular cooling, and application of extracorporeal assist devices (Appendix A). A waiver for written informed consent was obtained from the local ethics committee. The informed consent was permanently waived if the patient did not regain consciousness. Patients who regained consciousness were informed of their participation as soon as they were able to understand the purpose of the study. Post-resuscitation care was performed in accordance with the International Liaison Committee on resuscitation guidelines [23]. The primary outcome was neurologic function on day 30 after admission, which was assessed by independent study fellows blinded to levels of NET-related biomarker measurement. For primary outcome assessment, the five-point cerebral performance category (CPC) score was used, which classifies patients into good (CPC 1–2) and poor neurologic function (CPC 3–5; 3 = severe cerebral disability, 4 = coma or vegetative state, 5 = death) [24].

Resuscitation-related parameters were collected via structured telephone interviews with the dispatch center, the emergency physicians, and paramedics at the scene, as well as the bystander who made the emergency call. These parameters included location of cardiac arrest (home vs. public), initial rhythm (non-shockable vs. shockable), witness status, basic life support, downtime (interval from collapse to ROSC), the amount of epinephrine administered, and the administration of heparin by the emergency medical service (EMS). Demographics and chronic health conditions that existed pre-arrest were collected by review of past medical reports and interviews with relatives and the general practitioner if available.

The study was approved by the ethics committee of the Medical University of Vienna (EC Number 1674/2013) and conducted in accordance with Helsinki declarations.

### 2.1. Blood Sampling

Whole blood was obtained on admission immediately after vascular access was available and again 12 h later and stored in blood collection tubes containing 3.8% trisodium citrate (Greiner BioOne, Kremsmünster, Austria). Immediately thereafter, samples were centrifuged for 10 min at 3000 *g* and platelet poor plasma was stored at −80 °C until final analysis.

### 2.2. Laboratory Analysis of NET Related Biomarker

Citrullinated histone H3 (H3Cit), nucleosome, and cell free DNA (cfDNA) levels were obtained from plasma samples as previously described [12]. Briefly, cfDNA was measured using Quant-iT PicoGreen dsDNA Assay Kit (Thermo Fisher Scientific, Waltham, MA, USA) according to the manufacturer’s instructions. Nucleosomes were measured using Cell Death Detection ELISAPLUS (Roche Diagnostics, Mannheim, Germany) and the resultant values were compared to a plasma pool from male healthy controls. H3Cit levels were obtained by using a Cell Death Detection ELISA Kit (Sigma Aldrich, St. Louis, MO, USA). After overnight coating with anti-histone antibody at 4 °C, the 96-well plate (Nunc MicroWell 96-well microplates, Thermo Fisher Scientific, Waltham, MA, USA) was blocked with incubation buffer. After washing with phosphate buffered saline (PBS)-Tween, self-made H3Cit standards as well as plasma samples were incubated for 1.5 h at room temperature and washed again. Anti-H3Cit antibody (1:1000 ab5103, Abcam, Cambridge, MA, USA) was applied and incubated for 1.5 h at room temperature. After another washing step, secondary antibody (1:5000 goat anti-rabbit IgG horseradish peroxidase (HRP), Biorad, Hertfordshire, U.K.) was incubated for 1 h at room temperature and washed again. Incubation with TMB (3,3′, 5,5′-tetramethylbenzidine, Sigma Aldrich, St. Louis, MO, USA) for 25 min and the addition of 2% sulfuric acid resulted in a colorimetric change, readable at 450 nm. Measurement of NET biomarkers was performed without an awareness of neurologic assessment outcome. NET-related biomarkers were obtained in duplicate, and the respective mean value was used for the final statistical analysis. Resulted values are given in ng/mL, multiple-of-the-mean (MoM), and ng/mL for cfDNA, nucleosomes, and H3Cit respectively.

### 2.3. Statistical Methods

Categorical data are presented as absolute count numbers (*n*) and relative frequencies (%), continuous data as medians and 25–75% interquartile ranges. The patients were analyzed according to their neurologic function on day 30 (CPC 3–5/poor vs. 1–2/good). For between-group comparisons we used the Mann–Whitney U test for continuous variables and the Fisher’s exact test for categorical variables. We used a score test to assess a trend of increasing biomarker levels at specific time points for neurologic outcome and logistic regression models, including each relevant co-variable separately, to estimate the effect of NET biomarkers on neurologic function. A subgroup analysis of the effect of H3Cit on the primary outcome included only patients with a CPC of 1–4 on day 30 after admission. The score test is a nonparametric test for a trend across ordered groups as an extension of the Wilcoxon rank-sum test [25]. Results are given as crude odds ratio (OR) with 95% confidence interval (95% CI). NET biomarker levels were categorized into quintiles prior to analysis. The first quintile of each biomarker level distribution was used as a baseline reference. Covariables judged to be clinically plausible included age, sex, location of cardiac arrest (place of residence vs. public place), initial rhythm (non-shockable vs. shockable), witnessed status, basic life support, downtime (interval from collapse to ROSC equaling the sum of no-flow and low-flow time), amount of epinephrine administered, and d-dimer and lactate levels. D-dimer and lactate levels were log transformed to normalize data distribution. The likelihood ratio test was performed to assess deviations from linearity. The Spearman method was used to assess the correlation between plasma levels of cfDNA, nucleosomes, and H3cit. No data-imputation was applied for missing data. We used Stata Statistical Software (Release 14, StataCorp LLC, College Station, TX, USA) for data analysis and GraphPad Prism Version 8.0.2 for Windows (GraphPad Software, La Jolla, CA, USA) to draw figures. Generally, we considered a two-sided *p*-value < 0.05 as statistically significant.

## 3. Results

Between January 2014 and January 2017, 62 patients (79% male, median age: 57 years, 46–67) with OHCA who had achieved ROSC on admission were enrolled. In total, 52% of patients (*n* = 32) had a poor 30 day neurologic function. The number of patients with acute coronary syndrome was similar between patients with good and those with poor 30 day neurologic function (77% vs. 66%, *p* = 0.338). The time interval between collapse and study-related blood sampling was longer in patients with poor outcome (65 min vs. 56 min, *p* = 0.033). At 12 h, both the neutrophil count (11.8 vs. 9.9 G/L, *p* = 0.051) and the neutrophil-to-lymphocyte ratio (11.6 vs. 6.8, *p* = 0.046) were higher in patients with poor function compared with those with good neurologic function. The characteristics of the study patients, including median levels of NET-related biomarkers at 0 and 12 h, are shown in Table 1. Across all patients, there was no association between H3Cit and cfDNA (0 h: rho = 0.05, *p* = 0.718; 12 h: rho = 0.18, *p* = 0.189) or nucleosome levels (0 h: rho = 0.06, *p* = 0.669; 12 h: rho = 0.13, *p* = 0.328), neither on admission nor 12 h later. In contrast, cfDNA levels correlated with nucleosome levels at both time points (0 h: rho = 0.64, *p* < 0.001; 12 h: rho = 0.53; *p* < 0.001).

The poor neurologic function group had higher on-admission levels of cfDNA (1898 vs. 1197 ng/mL, *p* = 0.007) and nucleosomes (5.6 vs. 3.8 MoM, *p* = 0.032), but similar levels of H3Cit (434 vs. 447 ng/mL, *p* = 0.755) compared to patients with good neurologic function. While median levels of cfDNA and nucleosomes decreased in both groups from admission to 12 h, median levels of H3Cit increased in patients with poor 30 day neurologic function (Figure 1). At 12 h, all three biomarkers were higher in patients with poor 30 day neurologic function (cfDNA, 589 vs. 493 ng/mL, *p* = 0.016; nucleosomes, 1.1 vs. 0.4 MoM, *p* = 0.036; H3Cit, 667 vs. 299 ng/mL, *p* = 0.043). The score test showed a consistent trend towards poor neurologic function with increasing NET biomarker levels (*p* < 0.05) on admission (*p* > 0.85). This did not apply to H3Cit levels (Appendix A).

In crude regression analysis, the odds of poor neurologic function on day 30 increased with increasing levels of NET-related biomarkers at both time points (Figure 2). In this, 0 h levels of cfDNA and nucleosomes were associated with 1.8 (crude OR, 95% CI: 1.2–2.7, *p* = 0.007) and 1.7 (crude OR, 95% CI: 1.0–2.2, *p* = 0.049) times higher odds of poor neurologic function. 12 h levels of cfDNA, nucleosomes, and H3Cit were associated with 1.6 (crude OR, 95% CI: 1.1–2.4, *p* = 0.024), 1.7 (crude OR, 95% CI: 1.1–2.5; *p* = 0.02), and 1.6 (crude OR, 95% CI: 1.1–2.3; *p* = 0.029) times higher odds of poor neurologic function. In patients with a CPC of 1–4 (*n* = 50), the odds of poor neurologic function on day 30 likewise increased with increasing 12 h levels of H3Cit (crude OR 1.7, 95% CI 1.0–3.0). The test for deviation from linearity indicated a linear association for all these biomarkers. The effect remained unchanged after adjustment for covariables (Appendix A).

## 4. Discussion

Neurologic disability causes a high degree of morbidity in cardiac arrest survivors. This study investigated whether early plasma NET formation is associated with poor neurologic 30 day function following successful out-of-hospital resuscitation. The study was built on previous data suggesting a possible role of neutrophils in the development of post-resuscitative organ damage and was driven by the hypothesis that excessive NET release upon ischemic reperfusion may contribute to cerebral micro-circulatory compromise and thus neurologic disability in cardiac arrest survivors [5,6,7,8].

Previous studies reported increased mortality and neurologic morbidity in resuscitated cardiac arrest patients presenting with markers of neutrophil inflammation or elevated neutrophil counts in proportion to other leucocytes [6,7,8,26,27,28]. However, it has not been investigated whether these findings simply reflect stress response or whether neutrophils might be causally involved in the progression of organ injury following successful resuscitation.

This study provides a first indication that components of NETs may be involved in post-resuscitative brain damage. We found that NET-related biomarkers were already markedly elevated in cardiac arrest survivors at the time of admission, with these being even higher than levels measured in patients with cancer [12]. Although median levels of cfDNA and nucleosomes decreased over time, median levels of H3Cit increased in patients with poor neurologic function and were 30-fold higher at 12 h compared to those with good neurologic function on day 30.

cfDNA and nucleosomes are structural components of NETs but may have various sources and do not necessarily indicate their formation. Both are unspecific markers of cell death and cell turnover and may be interpreted as a measure of disease burden, but do not necessarily reflect the mechanism or source of disease. In this context, cfDNA levels have previously been shown to correlate with hospital mortality in cardiac arrest survivors [29]. H3Cit, in contrast, is considered a specific indicator of NET formation. Consistently, in this study, levels of cfDNA and nucleosomes on admission were significantly higher in patients with poor 30 day neurologic function, but decreased after successful resuscitation. In contrast, median H3Cit increased to significant levels at 12 h only in patients with poor 30 day neurologic function. It is likely that H3Cit are not detectable immediately upon admission, but take time to be formed, while less-specific markers originating from ischemic cell damage fall after ROSC. The lack of correlation between cfDNA and nucleosomes and H3Cit levels in our patients may further suggest that the biomarkers have different origins, and this is consistent with previous data on NET formation in patients with cancer [12], in whom H3Cit levels were likewise associated with an increased risk of mortality [30].

Although links in the chain of survival have substantially improved over the past decades, neurologic outcomes remain poor [31]. This is mainly attributable to the fact that several outcome-success factors cannot be influenced, such as witness status or a bystander’s capability to provide basic life support. It might, however, highlight the lack of knowledge of the mechanisms driving post-resuscitative organ damage, limiting current post-resuscitation care to targeted temperature management [32].

The potential of post-resuscitation care to improve neurologic outcome is yet to be realized. Timely targeted interventions may offer the opportunity to alleviate or even interrupt early organ damage cascades triggered by ischemic reperfusion. Our data show that NETs may be associated with poor neurologic function 30 days after successful resuscitation. In contrast to previously identified coagulation makers associated with poor outcome after cardiac arrest [33,34], NET components may have the potential to serve as therapeutic target structures. NET-targeting agents administered soon after or even during resuscitation could contribute to preventing secondary brain injury and improving neurologic function. Supporting evidence comes from a recent mouse model study that shows the critical involvement of high mobility group box 1 formation in NET formation [35]. High mobility group box 1 is released by neutrophils [36], and the inhibition of high mobility group box 1 formation upon ischemic reperfusion successfully attenuated post-resuscitative brain injury in a rat model study [37].

However, it remains to be determined which step of NET formation should be targeted for therapy and which component of NETs is most appropriate to serve as a target structure. From our data we are able to infer that the early inhibition of histone H3 citrullination by selective peptidylarginine deiminase 4 (PAD 4) inhibitors might be a promising approach. This has already proven effective in disrupting in vitro NET formation in mouse and human neutrophils as well as in vivo [38,39]. Therapeutic degradation of formed NETs by DNase 1 may be an alternative or additional approach, one that has already been successfully used in mouse models to prevent thromboembolic disease [40]. In septic mice, infusion of DNase 1 resulted in significantly lower quantities of intravascular thrombin activity, reduced platelet aggregation, and ultimately improved microvascular perfusion.

The use of DNase 1 might be particularly beneficial in cardiac arrest patients with myocardial infarction, as coronary DNase activity has been found to negatively correlate with coronary NET burden and also with infarct size [11]. In this study, we analyzed a group of patients who had cardiac arrest of cardiac etiology. These patients commonly receive early antithrombotic and anticoagulant therapy and may thus be prone to experiencing hemorrhagic complications. A possible advantage of PAD 4 inhibitors and DNase 1 may be that both are considered to preserve physiological coagulation and thus should not cause additional bleeding risk [40]. Until advancements are made in human clinical testing, we can only speculate on the safety of these substances in humans. If further studies confirm our results, however, interventional trials investigating the safety and efficacy of NET-targeting agents in resuscitated animals may be warranted.

### Limitations

The study was mainly limited by its sample size, resulting in large confidence intervals and the limited outcome events rate. We attempted to compensate for possible confounding by adjustment for clinically plausible covariables. Multivariable analyses including each relevant co-variable separately did not indicate relevant confounding. We included and analyzed a highly selected sample of patients with cardiac arrest of cardiac etiology who had achieved ROSC prior to hospital admission. Appropriate caution needs to be taken when interpreting our results. Furthermore, it must be mentioned that single patients with poor 30 day neurologic function did not show an increase in H3Cit levels from 0 to 12 h, while H3Cit levels did increase in some individuals with good neurologic outcome. Larger sampled studies investigating a more heterogeneous sample of cardiac arrest patients over a prolonged period of time may expand our results and aim at identifying specific patient characteristics, which allow for reliable prediction of NET formation in the individual patient in the early post-resuscitative stage.

Furthermore, a gold standard method for reliable measurement of NET formation in plasma is not available. We assessed three established NET-related biomarkers including H3Cit, which is considered a specific indicator of NET formation in plasma. It is, however, conceivable, that the assessment of additional NET components may gain test specificity and ultimately provide different results. The most reliable measure would perhaps be direct visualization of tissue NET formation, which may be investigated once a validated method becomes available.

Finally, we analyzed NET biomarker levels at only two time points, on admission and 12 h later. A more precise assessment of the time course of NET formation after cardiac arrest, however, may be of interest, because any NET-targeted therapy should likely be administered as early as possible to achieve the maximum beneficial effect.

## 5. Conclusions

We found that plasma levels of NET biomarkers assessed at an early post-resuscitative stage are associated with poor 30 day neurologic function in successfully resuscitated adults with OHCA. Further studies may assess whether NETs are causally involved in the pathophysiology of post-resuscitative brain damage, as well as discovering which components of NETs may have the potential to serve as therapeutic target structures to improve neurologic outcomes in cardiac arrest survivors in the future.

## Figures and Tables

**Figure 1 jcm-08-01568-f001:**
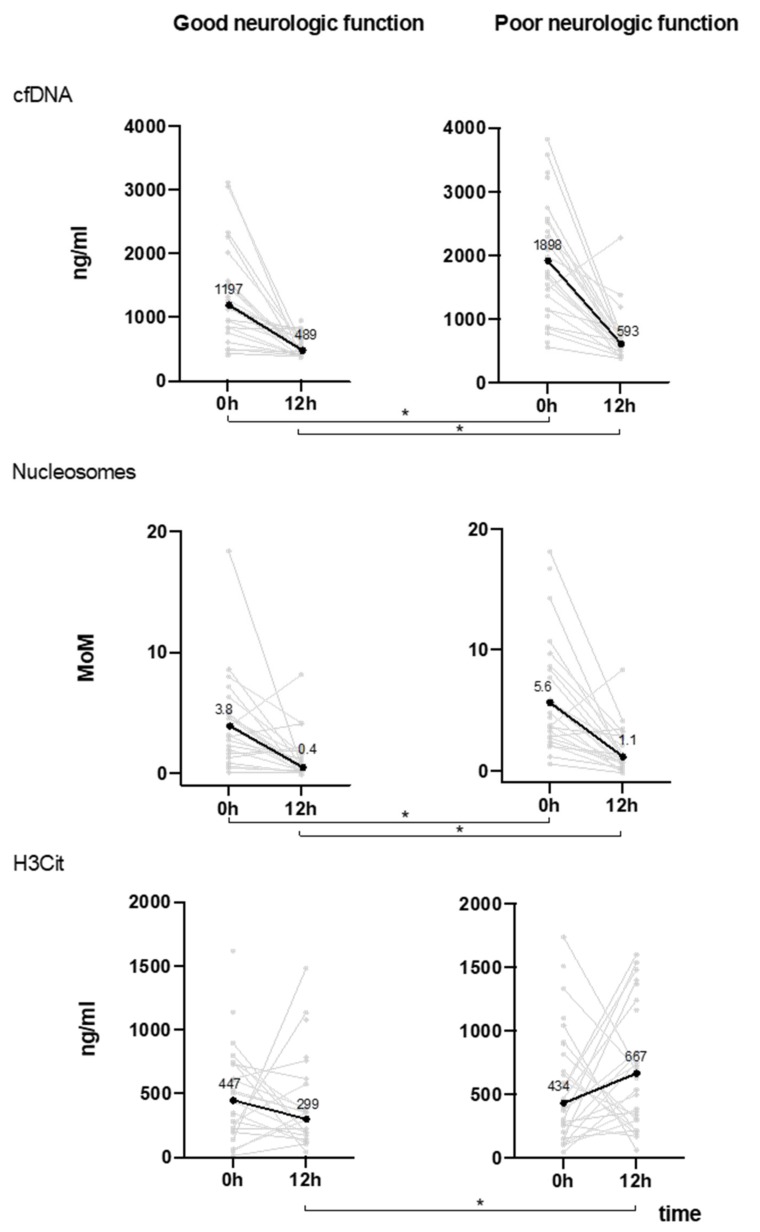
Plasma levels of neutrophil extracellular trap (NET) components (*y*-axis) on admission (0 h) and at 12 h (*x*-axis) in patients with good (left) and poor (right) 30 day neurologic function. Median levels of cfDNA and nucleosomes decreased in both groups from 0 to 12 h, while median H3Cit levels increased only in patients with poor 30 day neurologic function. Grey lines indicate individual data points, black lines represent median levels of NET-related biomarkers. * indicates significant difference. Individual and median d-dimer levels at 0 h and 12 h in the poor outcome group are available in the Appendix A.

**Figure 2 jcm-08-01568-f002:**
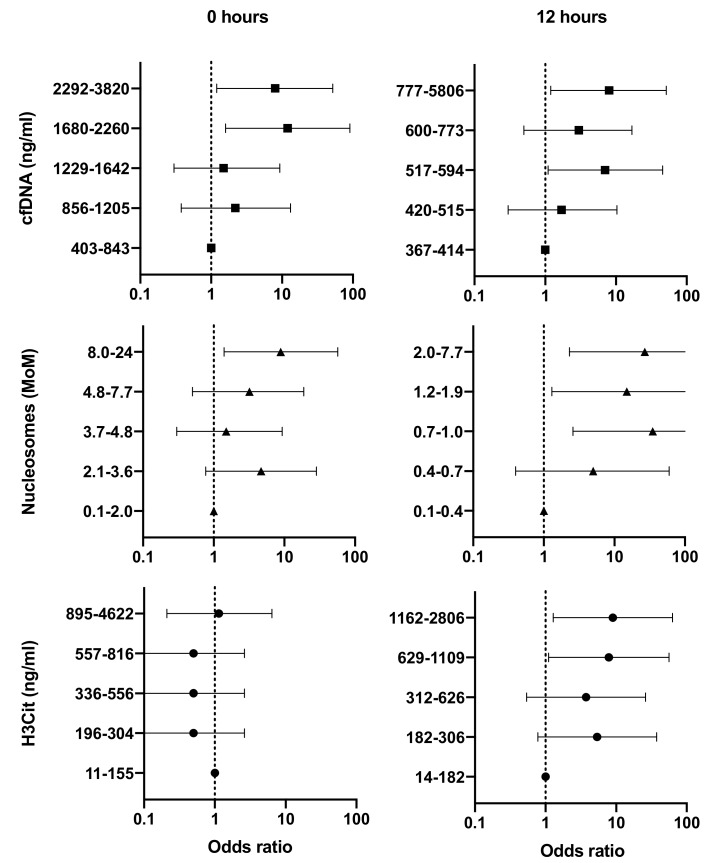
The crude odds of poor 30 day neurologic function (*x*-axis; crude odds ratio, log scale) according to plasma level quintiles of NET components (*y*-axis) on admission (left) and at 12 h (right). Confidence bands represent 95% confidence intervals. MoM, multiple-of-the-mean.

**Table 1 jcm-08-01568-t001:** Patient characteristics according to neurologic function on day 30. Data are *n* (%) or median (25–75% interquartile range).

Variable	Total (*n* = 62)	Good Function CPC 1–2 (*n* = 30)	Poor Function CPC 3–5 (*n* = 32)	*p*-Value
Male sex	49 (79)	25 (83)	24 (75)	0.421
Age, years	57 (46–67)	52 (44–61)	61 (53–70)	0.030 *
**Cause of cardiac arrest**				0.338
Acute coronary syndrome	44 (71)	23 (77)	21 (66)	
Primary arrhythmia	18 (29)	7 (23)	11 (34)	
PCI with stenting	42 (68)	22 (73)	20 (63)	0.362
**Resuscitation characteristics**				
CPC prior to cardiac arrest				1.0
CPC 1	61 (98)	30 (100)	31 (95)	
CPC 2	1 (2)	0	1 (5)	
*Location of cardiac arrest*				0.290
Place of residence	35 (57)	19 (63)	16 (50)	
Public place	27 (43)	11 (37)	16 (50)	
Witnessed	54 (87)	28 (93)	26 (81)	0.258
Basic life support	44 (71)	23 (77)	21 (66)	0.338
Shockable rhythm	46 (77)	27 (93)	19 (61)	0.004 *
Administration of heparin by EMS	28 (45.2)	14 (47)	14 (44)	0.795
Epinephrine, mg	3 (1–4)	1 (0–4)	3 (2–5)	0.004 *
Down time, min	29 (19–47)	23 (11–36)	38 (24–50)	0.011 *
Temp at admission, °C	35.3 (34.8–35.7)	35.4 (35.0–35.6)	35.1 (34.7–36)	0.676
Time from collapse to blood sampling, min	60 (49–72)	65 (52–87)	56 (43–65)	0.033 *
**Laboratory values**				
Lactate, mmol/L 0 h	7 (5–10)	6 (3–7)	10 (6–12)	0.001 *
D-dimer, µg/mL 0 h	8 (3–17)	4 (2–8)	14 (8–21)	0.003 *
D-dimer, µg/mL 12 h	4 (2–6)	2 (1–4)	6 (3–8)	0.009 *
Aptt, s 0 h	47 (36–121)	48 (33–129)	46 (37–119)	0.444
Aptt, s 12 h	37 (34–42)	37 (34–42)	38 (34–45)	0.487
Prothrombin time, % 0 h	79 (67–61)	79 (64–88)	78 (71–91)	0.418
Prothrombin time, % 12 h	77 (66–87)	80 (68–88)	76 (63–86)	0.549
Fibrinogen, mg/dL 0 h	290 (242–322)	297 (246–317)	283 (240–343)	0.983
Fibrinogen, mg/dL 12 h	297 (258–350)	295 (260–346)	308 (241–359)	0.502
Platelet count, G/L 0 h	204 (163–245)	204 (163–235)	204 (164–251)	0.972
Platelet count, G/L 12h	193 (141–240)	193 (141–239)	193 (150–252)	0.490
CRP, mg/dL 0 h	0.2 (0.1–0.6)	0.2 (0.1–0.4)	0.3 (0.1–0.7)	0.410
CRP, mg/dL 12 h	1.6 (0.7–3.1)	1.0 (0.3–2.8)	1.7 (1.0–3.3)	0.057
Neutrophils 0 h, G/L	8.5 (6.2–12.8)	7.8 (6.0–12.8)	9.6 (6.4–13.2)	0.443
Neutrophils 12 h, G/L	10.9 (8.5–14.6)	9.9 (7.7–12.2)	11.8 (8.8–16)	0.051
NLR 0 h	2.5 (1.4–4.4)	2.6 (1.4–4.7)	2.4 (1.3–3.8)	0.375
NLR 12 h	10.3 (6.1–14.8)	6.8 (5.7–11.5)	11.6 (7–18)	0.046 *
cfDNA 0h, ng/mL	1481 (948–2176)	1197 (835–1544)	1898 (1148–2377)	0.007 *
cfDNA 12 h, ng/mL	555 (436–721)	489 (404–634)	593 (516–807)	0.016 *
Nucleosomes 0 h, MoM	4.4 (2.4–7.1)	3.8 (1.6–4.9)	5.6 (2.8–9.7)	0.032 *
Nucleosomes 12 h, MoM	0.7 (0.3–1.8)	0.4 (0.2–1.1)	1.1 (0.5–2.4)	0.036 *
H3Cit 0 h, ng/mL	447 (228–772)	447 (229–744)	434 (205–899)	0.755
H3Cit 12 h, ng/mL	386 (207–968)	299 (146–789)	667 (300–1201)	0.047 *

cfDNA, cell-free DNA; CPC, cerebral performance category; CRP, C-reactive protein; EMS, emergency medical service; H3Cit, citrullinated histones H3; MoM, multiple-of-the-mean; NLR, neutrophil-to-lymphocyte ratio; PCI, percutaneous coronary intervention; Temp, temperature; TnT, troponin. * indicates significance.

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
