# Peer review of "Increased Citrullinated Histone H3 Levels in the Early Post-Resuscitative Period Are Associated with Poor Neurologic Function in Cardiac Arrest Survivors—A Prospective Observational Study"

_jcm, 2019, doi:10.3390/jcm8101568_

Round 1

Reviewer 1 Report

The present study by Mauracher et al. investigates the utility of citrullinated histone H3 levels in cardiac arrest patients. The Authors conclude that the level of citrullinated histone H3 may be an important biomarker to predict a decline in neurological function in this patient population. Although the topic is of interest and the data is well presented i have a number of concerns .

1)  No information is provided regarding any potential causes of the OHCA. It would, for example, be interesting, to know how many patients had atrial fibrillation, or other such disorders.  Furthermore, there is no information regarding their coagulation status at the time points that blood was taken. Given the fact that the Authors have previously published on the relationship between this biomarker and thrombus formation, i believe these data are vital.

2) Is there any information regarding the timing between occurrence of OHCA and the first blood sampling?

3) I think it would be good to show significance values on Table 1 between good and poor neurological outcomes. This would make it easier for the reader of the article.

4) When I look at Figure 1 i notice there are a number of patients without 12 hour data. I'm assuming these were patients who died? I am therefore not too sure how useful it is to include these patients, as their neurological outcomes will also not have been able to have been measured. Are these included in the n numbers? Therefore, I believe there needs to be more discussion of this. Also the legend for this figure needs to be rewritten to describe exactly what is shown- for instance I'm assuming the light grey lines are individual data points, and the darkened (or red) line is the median patient, however I cannot be sure of this.

5) I also think it could be interested to look at the % of patients that showed an increase or decrease with the different biomarkers over the 12 hour period. It is stated that 'H3Cit levels increased only in patients with poor 30-day neurological recovery' however this is not true, as there are a number of patients in the good recovery group where the levels also increased. Therefore, the Authors need to be careful not to overstate the findings of the study.

6) I believe there to be some repetition in the methods regarding obtaining of the waiver forms.

7) It should also be stated in the methods when the neurological outcome measurements were obtained.

Reviewer 2 Report

Abstract

Abstract should be an independent writing in itself. Authors mentions specific and putative biomarker at 0 and 12 h after admission in the method. However, authors show only the analysis of specific biomarker at 12 h after admission in the result.

Line 16-17: It would be better to state the aim of this study clear.

Line 16-17: return of spontaneous circulation (ROSC) is more common expression rather than “after successful resuscitation” in cardiac arrest research.

Line 18: adult patient with OHCA of cardiac origin can be used as adult OHCA survivor with cardiac origin.

Line 19: adult OHCA survivor has the meaning of achieving ROSC

Line 23: Please state clear what is the primary outcome and the primary outcome should be defined. State how to measure the neurologic outcome. Terms should be unified (neurologic impairment or poor neurologic outcome)

Line 25. “1.1-2.3” looks confidence interval. It should be identified.

Line 26. Authors mentioned that fifth quintile had highest odds ratio. Authors need to let reads know what is the baseline reference and whether other quintile had significant odds ratio.

Introduction

Line 46-47: It would be better to show the citations.

Line 58-60. Authors intended to show the association between NET biomarker and neurologic outcome. The meaning of sentence overlaps and lengthens. This paragraph would be better to describe author’s hypothesis and aim.

Line 60: unfavorable; authors need to unify the term “unfavorable” or “poor”

H3Cit is the main result of this study. However, there is no introduction of H3Cit.

Line 61. Methods is appropriate rather than “Experimental section”

Line 63. Adult should be defined with what age.

Line 64. Return of spontaneous circulation can be used as an abbreviation of “ROSC”. It is common abbreviation in cardiac arrest research field.

Line 64-65. Did authors include only the patient who achieved prehospital ROSC? There is no description of exclusion criteria.

Line 65. “A waiver…if they regained consciousness” is duplicated in Line 72. The meaning of this sentence is not clear. What waiver was obtained and who waived? Informed consent? The informed consent was permanently waived if the patient did not regain consciousness? Please state clear.

Line 68-70. The word order is not appropriate.

Line 76. Is “Greiner BioOne, Kremsm?nster, Austria” the company of EDTA tube?. It would be better to express like “Whole blood was obtained stored in EDTA tube (Greiner BioOne…)”

Line 103: Authors seem to perform the score test for trend with quintile of each NET biomarker for poor neurologic outcome. However, explanation for this analysis is not enough.

Line 106-108: How authors did obtain cardiac arrest characteristics such as location of cardiac arrest, rhythm, witness, no-flow time, low-flow time etc. The clinical variables which were obtained should be mentioned and defined if needed. For example, I guess that reader cannot get the meaning of “preclinical heparin”.

Line 111. Correlation between H3cit, necleosome, cfDNS seemed provided in the results. Then, please state exactly rather than Net biomarkers.

Results

Although this is a prospective study, this study does not provide flow diagram with exclusion criteria. How many OHCA was screen and how many OHCA was excluded from this research with some reason?

Line 123, Please provided p values with rho

Table 1: Does downtime mean “no-flow interval + low-flow interval”? It would be better unify the variables in Methods and Results.

Table 1. Foot note should be place under the table.

Table 1. How was the result of comparisons between good and poor outcome groups? Those were all insignificant?

Line 130: Patients with poor 30-day neurologic outcome group can be changed to “Poor neurologic outcome group” in short.

Line 135. “Highest” is inappropriate, since authors compared two groups.

Line 136-138.: The score test result should be organized in Table S1. The result of score test is not a Table.

Line 136-138. Except H3Cit on admission, Poor neurologic outcome group had higher nucleosome and cfDNS levels on admission than good neurologic outcome. As considering higher level of those NET level on admission, is it appropriate to say that nucleosome and cfDNA at 12 h after admission have a increasing trend in poor neurologic outcome group. It would be helpful to analyze the difference between at 12 hr and on admission levels to interpret the data in depth. Authors seemed to miss the effect of interaction of time between 0 and 12 h after admission.

Line 144. “Neurologic impairment” means unclear. Unify the terms

Line 144 and Figure 2. It looks the result of multivariate logistic regression. If so, state clearly, they are adjusted OR. Although authors mentioned in method section, please add in footnote which variables were used for adjustment. If not, state clearly they are Crude ORs.

Discussion

Line 157:”previous data” Citation needed

Line 163-165: What did authors developed with this study in this point view?

Line 166-167: Neurologic outcome after cardiac arrest depends on not only brain damage but also systemic organ dysfunction, even if the patient achieved ROSC. Therefore, this sentence seems to a conclusion beyond the result of this study. To elucidate the association between NET increase with brain damage, NET should higher

Reference

Should be corrected according to Journal guidelines

Reviewer 3 Report

The present study of Mauracher et al deals with the interesting topic of inflammation and biomarkers in cardiac arrest. Based of some previous published data of NETs the authors focused on their effect in CA.The manuscript is well written. However some comments need to be addressed:

The study cohort is classified as "... with out-of-hospital cardiac arrest of Cardiac Etiology". This "Cardiac Etiology" needs to be defined and described in detail (primarily arrhythmia or ischemic heart disease / acute MI) since elevated NETs have been described in patients suffering from STEMI and have been correlated with infarct size (Hofbauer TM et al 2019, Basic Res Cardiol). NETs may be more a marker of arterial thrombembolism which is already described. Subgroup analyses will be difficult due to limited sample number.

Since inflammation is the superior topic when NETs are discussed, the authors should provide some additional data of labwork (like C-reactive protein and maybe procalcitonin). What about the infections (pneumonia) ? Is the worse outcome in patients with increased H3Cit levels dependent on infectious complications like pneumonia and sepsis?

What about the effects of sample preparation. Was centrifugation and storage in -80° immediately done following whole blood collection?

Are there any data available of H3Cit levels in healthy controls, isolated by the same procedure as the study cohort (regardless of body temperature)?

Round 2

Reviewer 1 Report

I thank the Authors for dealing with my previous comments. I believe the revised manuscript is much improved. However, i still have a number of comments/questions.

1) Now the information regarding D-Dimer is shown in more detail, it appears that the levels of this bio-marker is also linked to the neurological outcome. What, therefore, is the benefit of the novel marker? What would be the odds-ratio with d-dimer, for example? There should be some discussion regarding this.  I think an additional analysis that would be of interest would be to plot d-dimer levels against against the levels of citrullinated histone H3 in individual patients.

2) Please check for spelling/grammar throughout the manuscript. There are a number of typos, eg 'ad' instead of 'as' on line 144

3) I believe the data when only patients that survive are included given in the reply to reviewers should be included in the text of the manuscript. 

Reviewer 2 Report

Thank the authors for their appropriate responses to my review for massive changes made to the paper.

The changes make sense.

I found some typo or error.

Line 24: Please add “cfDNA” in parentheses

Line 39: I guess that “lower quintiles” should be changed to “the lowest quintile” or “the first quintile”.

Line 97: I think Resuscitation-related parameters were “collected”, not “assessed”.

Line 144” given “as”, not “ad”

Line 159 and line 311: out-of-hospital cardiac arrest can be “OHCA”

Reviewer 3 Report

The authors nicely and intensively took care for my comments.
